# Large Language Model is not a (Multilingual) Compositional Relation Reasoner

**Jinman Zhao** [*][†]
Department of Computer Science
University of Toronto
Toronto, Canada
jzhao@cs.toronto.edu

**Xueyan Zhang** [*]
Department of Computer Science
University of Waterloo
Waterloo, Canada
xueyan.zhang@uwaterloo.ca

## Abstract

We present a comprehensive evaluation of large language models' capability to reason compositional relations through Multilingual Compositional Relation (MCR) Benchmark in both English and Chinese, covering six distinct categories of compositional relations: Positional, Comparative, Personal, Mathematical, Identity, and Other. We expand our assessment to the multilingual realm by including translations of the benchmark suite into Japanese, French, and Korean. Our MCR aims at investigating the robustness and adaptability of LLMs in handling compositional relation reasoning across diverse linguistic contexts. [1].

## 1 Introduction

Since the emergence of LLMs (Brown et al., 2020; Touvron et al., 2023; Chowdhery et al., 2023), there has been a heightened focus on their reasoning capabilities. Notable research efforts (Wei et al., 2022; Yao et al., 2023; Hendrycks et al., 2021; Hu et al., 2024; He et al., 2024; Viswanathan et al., 2023) focused on evaluating LLMs' abilities in various dimensions of reasoning. These capabilities encompass a wide range of cognitive skills, including but not limited to, arithmetic, commonsense and symbolic reasoning. Such comprehensive scrutiny aims to uncover the extent to which these advanced models can mimic, if not surpass, human-like reasoning processes across diverse scenarios and complex problem-solving tasks.

The concept of compositional relations extends far beyond its mathematical concepts, where, for instance, a composition function represents a specific type of compositional relation. In Natural Language Processing (NLP), compositional relations are essential for several reasons. 1) It enables sophisticated understanding and generation of language by allowing models to discern and construct complex relationships between entities within a sentence or across texts. For instance, understanding that *"uncle"* refers to a compositional, familial relation since it means *"one's parent's brother"*. So it is a compositional relation of *"brother of"* and *"parent of"*. 2) Leveraging compositional relations enhances the ability of NLP systems to handle ambiguity, infer missing information, and grasp the underlying semantics of text, leading to more effective in processing and better language coherence.

Natural Language Understanding (NLU) (Storks et al., 2019; Wong et al., 2023) is a sub-field of NLP focused on enabling machines to understand and interpret human language in a way that is both meaningful and contextually relevant. Relation Extraction (Pawar et al., 2017; Smirnova & Cudré-Mauroux, 2018) is a fundamental component of NLU that involves identifying semantic relationships between entities mentioned in the text. Compared to classic benchmarks in Relation Extraction (Han et al., 2018; Yao et al., 2019; Zhang

---

[*]Equal contribution
[†]Corresponding author
[1]Our benchmark MCR is released at https://github.com/zhaojinm/Multilingual_Composition_Relation_benchmark

et al., 2017), our benchmark places a greater emphasis on the **composition** aspect. For example, instead of deducing the relationship *"James is educated at Liverpool University"* from a detailed sentence like *"James Alty obtained a 1st class honors (Physics) at Liverpool University."* Our benchmark consists of direct and concise statements; we mainly focus on deriving relationships similar to inferring the size relation and comparison between "Star A" and "Star C" from a statement, *"Star A is larger than Star B, and Star B is larger than Star C."*

The reasoning capabilities of LLMs in English have been extensively studied and analyzed Huang & Chang (2022). We expand our scope to evaluate the multilingual reasoning abilities regarding compositional relations, through the selection of languages that are topologically diverse and span a variety of language families. French, Japanese, and Korean were selected to represent a diverse linguistic spectrum, challenging the models' versatility across different linguistic structures and idioms.

We summarize our main contributions below:

- We develop a bilingual benchmark to assess the capability of LLMs in reasoning about compositional relations.
- Upon testing various LLMs on MCR, we observed discrepancies between the LLMs' reasoning and human reasoning. The performance of some models is worse than that of random guessing by chance.
- We extend our evaluation to include a multilingual aspect.

## 2 Related Work

**(Multilingual) benchmarks**  Although LLM can generate data itself (Zhao et al., 2024), people still rely on human-labeled benchmarks to evaluate LLMs. Many recent benchmarks are designed to evaluate the fundamental reasoning capabilities of LLMs, focusing on areas such as commonsense understanding, symbolic reasoning and arithmetic proficiency. For example, CommonsenseQA (Talmor et al., 2019) and StrategyQA (Geva et al., 2021) are created for commonsense reasoning. Last Letter (Wei et al., 2022), BigBench Date (Suzgun et al., 2023) and Coin Flip (Wei et al., 2022) are designed for symbolic reasoning. There are some benchmarks for arithmetic (Cobbe et al., 2021; Ling et al., 2017; Patel et al., 2021), temporal reasoning (Su et al., 2024), coding (Tang et al., 2024). All these benchmarks differ from ours, as our benchmark encompasses mathematical operations and logical reasoning. Our emphasis is more on assessing the LLMs' comprehension of understanding languages.

For the multilingual aspect, MMLU (Hendrycks et al., 2021) emphasizes multilingual capability, assessing models on their ability to handle tasks in various languages. XQA Liu et al. (2019) and MLQA (Lewis et al., 2020) are novel datasets of cross-lingual question answering. XNLI (Conneau et al., 2018) is a cross-lingual Inference corpus by extending NLI corpora to 15 languages. MGSM (Shi et al., 2023b) is a multilingual arithmetic benchmark that extended from GSM8k.

**Large Language Models are inconsistent with humans**  Some recent works show LLMs have different behaviors than humans. Berglund et al. (2024) demonstrates the failure of auto-regressive LLMs to learn the reverse relation that "B is A" by training on "A is B". Grosse et al. (2023) has the similar result that observed that training examples like "A precedes B" had a significantly greater impact compared to examples where the order was reversed, i.e., "B precedes A." Ullman (2023) shows LLMs often fail when subjected to even minor alterations in tasks involving the theory of mind, casting doubt on LLMs' ability to replicate human-like reasoning. Huang et al. (2023) primarily investigates the self-correction capabilities of LLMs, unveiling both their potential and limitations. This research reveals that in terms of reasoning, LLMs struggle to self-correct without external feedback, and occasionally, their performance may even deteriorate after attempts at self-correction. Shi et al. (2023a) point out that LLMs are easily distracted by irrelevant information during reasoning. Shanahan et al. (2023) illustrate LLM-based dialogue agents are not conscious beings with personal motives or a sense of self-preservation; their display of such traits is simply an act of role-play.

## 3   Motivation

If a LLM has the ability to predict a **definition** through complex relationships, such as recognizing that:

*"father's father is known as the grandfather."*

where the LLM's prediction is underlined. Meanwhile, given *"A is B's son"*, LLM successfully predicts the backward relation, *"B is A's father"*; then, it logically follows that the model should be able to infer complex familial relationships from the information provided. For instance, in the following relation implication,

*"A is B's son, and B is C's son."*

It would be a natural conclusion, aligning with human cognitive reasoning, to deduce that

*"C is A's grandfather"* or *"A is C's grandchild"*.

The inability of LLMs to make such inferences is considered as one of the many road-blocking limitations, revealing a gap between the basic pattern recognition and the deeper, contextual understanding that underpins human cognitive processes. This inconsistency not only undermines the intuitive understanding of relationships but also casts doubt on the model's claim of meaningful comprehension.

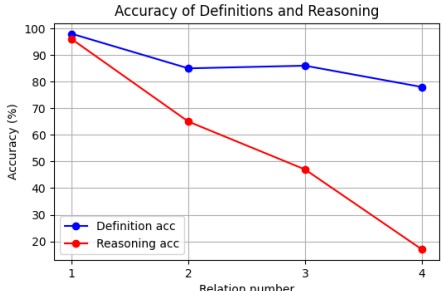

| Category | Size |
|---|---|
| Positional | 347 |
| Comparative | 325 |
| Personal | 333 |
| Mathematical | 326 |
| Identity | 242 |
| Other | 307 |

Figure 1: The performance of ChatGPT across the two types of questions.

Table 1: Statistical information from MCR benchmark for each language.

To contrast the LLM's performance difference between **definition** and **reasoning** task, we crafted 100 intricate family relationship queries, as shown above, for 4 levels of relational complexity, which we refer to as **reasoning** questions. We refer *"A is B's son, B's father is?"* as a single relation and *"A is B's son, B is C's son, C's grandchild is?"* as two composition relations reasoning, and so on. Alongside each reasoning question, we formulated a corresponding **definition** question, like *"Is the father's father termed as grandfather?"*. That leads to 800 questions in this experiment. It's important to note that in this study, we also opt to use Chinese, attributing to the language's greater precision in describing familial relationships compared to English. For example, English uses the single term "grandfather" to denote both "father's father" and "mother's father," whereas Chinese differentiates these relationships as "爷爷" and "外公" respectively. Figure 1 presents the accuracy metrics generated by ChatGPT, highlighting a discernible gap between the model's performance on **definition** interpretation and **relation** inference. It is worth mentioning that the drop in accuracy is markedly more pronounced in complex relational reasoning than in drawing inferences from definitions. This discrepancy raises concerns about whether LLMs, despite their advanced capabilities in predicting the next token, genuinely possess a deep understanding or are merely sophisticated pattern recognizers. Additional examples can be found in Appendix A.1 and Appendix A.2.

# 4 Benchmark

The objective of our research is to investigate LLMs' capability of reasoning through the compositional relations from a set of self-contained questions. Given two relations $R$ and $S$, the composition of $R$ and $S$, denoted as $R \circ S$, is defined as the set of ordered pairs $(a, c)$ such that there exists an element $b$ where $(a, b) \in R$ and $(b, c) \in S$. Formally, this can be expressed as:

$$R \circ S = \{(a, c) \mid \exists b \, ((a, b) \in R \wedge (b, c) \in S)\}$$

It denotes that for any $a$ and $c$, $a$ is related to $c$ through the composition $R \circ S$ if and only if there is an intermediate $b$ such that $a$ is related to $b$ through $R$ and $b$ is related to $c$ through $S$. Table 1 depicts the statistics of our MCR Benchmark.

We investigate such compositional relations by providing a LLM with a prompt $p$ containing $a$ to $b$ and $b$ to $c$, and we then measure the probability of generating $a$ to $c$ as a response. The prompt $p$ is in the form of Multiple Choices Question (MCQ) with 3-6 choices labeled with an uppercase letter starting from A. Each MCQ only has one most appropriate answer. If the likelihood of a model failed to choose the most appropriate answer, then the model is considered less effective in compositional relation reasoning.

## 4.1 Benchmark Creation

In this section, we present the construction and translation process of Multilingual Compositional Relation (MCR) Benchmark, the first multilingual compositional relation benchmark to the best of the authors' knowledge at the time of writing.

**Benchmark Construction**   We construct dataset made up of compositional relation questions with a set of possible options in both English and Chinese. Both are reviewed by native speakers. Each question in the dataset includes two or more compositional relations, $|R| \geq 2$. Given the relations described in the prompt $p$, there is one and only one appropriate option inferred as a response.

These questions are specially engineered so that humans can easily answer them with commonsense knowledge and fundamental mathematics. All questions are self-contained, indicating that the context for all applicable relations for deciding among available options are provided. Given the intention of this work is to investigate state-of-the-art LLM's reasoning capability in compositional relations, we do not employ any fine-tuning process.

**Benchmark Validation**   We have employed a group of graduate students, including authors, and divided them into three distinct roles: question creators, answer verifiers and quality checkers. In the question-creating stage, we sourced texts from various online platforms, such as social media and Wikipedia, and then gathered simple relationships within these texts. For example, comparative relationships like *"[entity A] is more [adj] than [entity B]"* were patterns we searched for. We then trained a group of seven workers to integrate different relationships and rephrase them into questions.

For the verification step, we assigned four workers as answer verifiers. Each question was distributed to two workers, and the question would be removed if both workers answered it incorrectly. This procedure helped us filter out roughly 10% of the initial questions, ensuring a higher level of accuracy and a lower level of ambiguity. Lastly, we trained two workers to assess the quality of both the questions and their corresponding answers. These two quality checkers had access to the complete information, focusing particularly on detecting any potential ambiguities in the questions.

## 4.2 The Task

Despite personal relationships, our benchmark also encompasses five other types of relationships that are commonly studied by other research (Jindal & Liu, 2006; Jie et al., 2022;

**Example 1: Comparative Relation**

Air conditioning was invented one year earlier than airplanes, airplanes were invented one year earlier than tractors. When was air conditioning invented compared to tractors?

A. One year earlier

B. Two years earlier

C. One year later

D. The same year

Answer: B. Two years earlier

(a) Comparative Relation

**Example 2: Identity Relation**

In a certain region, April's rainfall is 5 milLilyters more than May's, and June's rainfall is 5 milLilyters more than May's. How does April's rainfall compare to June's?

A. 10 milLilyters less

B. 10 milLilyters more

C. The same amount

D. Uncertain

Answer: C. The same amount

(b) Identity Relation

**Example 3: Mathematical Relation**

In a plane, line AB is parallel to line CD, and line CD is parallel to line EF. What is the relationship between line AB and line EF?

A. Parallel

B. Perpendicular

C. Intersect

D. Uncertain

Answer: A. Parallel

(c) Mathematical Relation

**Example 4: Personal Relation**

Mike is William's grandfather, Mary is William's wife, May is Mary's daughter. May might be William's ?

A. Aunt

B. Daughter

C. Sister

D. Grandmother

Answer: A. Aunt

(d) Personal Relation

**Example 5: Positional Relation**

Star A is in the northeast direction of Star B, and Star C is north of Star A. In which direction is Star C from Star B?

A. East

B. North

C. Uncertain

D. Northeast

Answer: D. Northeast

(e) Positional Relation

**Example 6: Other Reasoning Relation**

Williams participates in the Miss Universe contest, one of the selection criteria for which is being enthusiastic. What kind of person is Williams?

A. Positive

B. Enthusiastic

C. Beautiful

D. Uncertain

Answer: B. Enthusiastic

(f) Other Reasoning Relation

Table 2: Example Questions for each category in MCR Dataset. Notice that a question may fall into one or more categories.

Weston et al., 2015). Example questions for each category is shown in Table 2. For more examples, please refer to Appendix B.

**Comparative Relation**   In this task, our primary focus is to test complex, multi-level comparative relations. We present comparative relationships among several objects and then query the comparative relation between any two of those objects, as demonstrated in Table 2a.

**Position Relation**   This category of questions gives the positional relationships among several objects, followed by a query regarding the positional relationship between two specific objects. Table 2e provides an example of such a question.

**Mathematical Relation**   This category of questions will give various mathematical relations, followed by querying the relationship between two objects. Our benchmark encompasses questions related to geometry, algebra, and arithmetic. Table 2c depicts an example of a mathematical relation.

**Identity Relation**   This task involves presenting the relation between various objects, ultimately revealing that two objects share identical characteristics. Identity relation questions usually do not appear standalone, and may also be classified under other categories of questions. For instance, the question illustrated in Table 2b is also classified as the comparative.

**Personal Relation**  This category of questions is used as examples in the introduction sections and they typically involve complex interpersonal titles reasoning, as demonstrated in Section 3 and Table 2d.

**Other Reasoning Relation**  This type of question contains various logical relations. For an illustration, refer to Table 2f.

# 5  Experiment

We conduct a series of experiments to evaluate the capability of state-of-the-art LLMs in reasoning compositional relation questions with various prompt settings in different languages. In this section, we will first list out the all LLMs investigated, introduce the prompting settings, and then present an analysis of the results in MCR.

## 5.1  Experimental Setup

**Model Selection**  We use the following commercially-available and open-sourced large language models:

- GPT-3  (Brown et al., 2020) was widely adopted and tested across abundant benchmarks in popular literature. Specifically, we use **text-davinci-002**.
- ChatGPT is the most popular, proficient and economically efficient model within the GPT-3.5 (Ouyang et al., 2022). We use **gpt-3.5-turbo-1106**.
- GPT-4 [2] is a multi-modal that surpasses all its predecessors in GPT families. Specifically, we use **gpt-4-0613**.
- Llama2 (Touvron et al., 2023) 7B/13B Chat model which are open-sourced decoder-only LLM. We use the **Llama-2-7b-chat-hf** [3] and **Llama-2-13b-chat-hf** [4].
- Mistral-7b offered by Mistral AI is a pretrained generative text model, claimed to outperform language models of its size in extensive benchmarks. Specifically, we use **mistral-7b-instruct-v0.2** [5] checkpoint.

In MCR, all questions are multiple choices; thus, in all our experiments, we employ greedy decoder sampling by setting the temperature to 0.

**Prompting techniques**  In our assessment, the prompt we use follows the recent LLM QA prompting research(Wei et al., 2022; Kojima et al., 2022; Xu et al., 2024; Shi et al., 2023b). We thoroughly assess how well the LLM performs using widely adopted prompting techniques.

- Zero-shot (ZS) prompt. *"Q:{Input query}. Choices: {Options}. A:"*
- Few-shot prompt. QA format with a few exemplars.
- Zero-shot chain-of-thoughts (ZSC) (Kojima et al., 2022) prompt. *"Q:{Input query}. Choices: {Options}. A: Let us think step by step."*

**Target Language Translation.**  We select 3 typologically distinct languages besides English (EN) and Chinese (ZH), spanning various language families and different levels of representation in common LLM training datasets, including French (FR), Korean (KO) and Japanese (JA). In contrast to the experiment conducted by Shi et al. (2023b), which investigates the reasoning capabilities facilitated by translating from various languages into English, our current study explores the inverse process: translating from English into other languages. We employ Google Cloud Translate API for two reasons. First, due to its well-documented

---

[2]https://platform.openai.com/docs/models/gpt-4-and-gpt-4-turbo
[3]https://huggingface.co/meta-llama/Llama-2-7b-hf
[4]https://huggingface.co/meta-llama/Llama-2-13b-hf
[5]https://huggingface.co/mistralai/Mistral-7B-Instruct-v0.2

proficiency in empirical machine translation outcomes, as highlighted in the existing literature (Zhu et al., 2023). Secondly, there is a potential for bias if we were to translate with the same LLM that we employed for evaluating our benchmark. This concern arises from the possibility that the LLM could inherently favor its own methodology or underlying data.

| "Tom is 300 meters east of the teaching building, and the teaching building is 500 meters south of Lily. Where is Tom in relation to Lily?

Answer choices: A) Northeast B) Same Location C) Southeast D) Southwest

*Let's think step by step.*" | "Dans une certaine force terrestre, le nombre de chars est le double de celui des véhicules blindés, et le nombre de missiles sol-air est le double de celui des chars. Combien de missiles sol-air y a-t-il par rapport aux véhicules blindés ?
Answer choices: A) Quatre fois plus B) Six fois plus C) La même quantité D) Incertain

*Let's think step by step.*" | "苹果MacBook Pro的处理器性能比戴尔XPS的处理器性能高,联想ThinkPad的处理器性能比微软Surface的处理器性能低,苹果MacBook Pro的处理器性能与微软Surface的处理器性能相同,那么联想ThinkPad的处理器性能与戴尔XPS的处理器性能比较如何?
Answer choices: A) 高B) 低C) 一样D) 不确定

*Let's think step by step.*" |
|---|---|---|
| (a) English (EN) | (b) French (FR) | (c) Chinese (ZH) |

Table 3: **Zero-shot Chain-of-Thought Prompting Example.** EN-COT represents prompting the model to reason in English despite the problem language. EN-COT is adopted for all Chain-of-Thought experiments in this paper.

**CoT Prompting Language Selection** In a multilingual environment, there are various combinations available when it comes to standard Chain-of Thought (CoT) prompting. Intuitively, for a given target language we can always adopt the same language in the zero-shot or few-shot prompts, which is referred as NATIVE-COT (Shi et al., 2023b). An alternative is to consistently prompt the model in English (Hu et al., 2020; Zhao & Schütze, 2021), despite of the target language, which is known as EN-COT.

Building upon the evidence provided by prior research (Shi et al., 2023b; Wei et al., 2022; Schick & Schütze, 2021; Winata et al., 2021; Lin et al., 2022), it has been observed that EN-COT consistently achieves marginally higher accuracy than NATIVE-COT across various scenarios. As CoT prompt language goes beyond the scope of this study, we thus have opted to adopt EN-COT prompting as default. That is, the questions in the MCR benchmark are translated into target languages, and when it comes to Zero-shot chain-of-thoughts (ZSC) prompting, English is used in the rest of the prompt, as shown in Table 3.

## 5.2 Main Results

Table 4 presents the accuracy achieved by six state-of-the-art LLMs on Multilingual Compositional Relation (MCR) Benchmark using zero-shot (ZS), few-shot (FS), and zero-shot chain-of-thought (ZSC) prompting settings described in Section 5.1.

**Effect of Models** Overall, it is clear that GPT-4 outperforms the other models by a substantial margin in both ZS and ZSC scenarios across all languages. Notably, GPT-4's ZSC performance averages at 67.2%, surpassing the second best, ChatGPT by approximately 25%. GPT-4 demonstrates an accuracy nearly three times higher than that of the least performant model, Llama2 7B. This suggests that GPT-4's advanced architecture and humongous training data afford it superior comprehension and reasoning capabilities in compositional relations. Given its parameter size of 7 billion, Mistral 7B achieves an impressive, commendable performance in MCR, achieving results that were nearly on par with ChatGPT, albeit slightly lower; it surpasses the other two open-source models, Llama 2-7B and Llama 2-13B, conforming to findings in Jiang et al. (2023). Llama 2-13B outperforms GPT-3 and Llama 2-7B by a noticeable margin. The comparatively lower accuracy of GPT-3 and Llama 2-7B, which approaches the level of random guessing, indicates that both architectural differences and variations in training data may significantly influence performance outcomes.

**Effect of Prompting** For all models, the average accuracy in the zero-shot chain-of-thought (ZSC) is noticeably higher than in the standard zero-shot (ZS) methodology. With a clear

|  | Prompt | AVG | EN | FR | JA | KO | ZH |
|---|---|---|---|---|---|---|---|
| | ZS | 21.78 | 22.47 | **23.15** | 22.47 | 20.03 | 20.79 |
| Llama2 7B | 5-shot | 25.06 | 22.15 | 26.85 | 24.34 | 24.03 | **27.91** |
| | ZSC | 24.54 | **27.53** | 25.16 | 22.78 | 21.84 | 25.39 |
| | ZS | 27.61 | 28.91 | 28.41 | **29.29** | 23.28 | 28.17 |
| Llama2 13B | 5-shot | 27.89 | **30.79** | 26.10 | 29.66 | 27.78 | 25.14 |
| | ZSC | 29.15 | 30.35 | **31.66** | 27.97 | 26.60 | 29.17 |
| | ZS | 29.33 | **32.67** | 30.98 | 26.85 | 27.97 | 28.17 |
| Mistral 7B | 5-shot | 31.04 | **34.79** | 30.85 | 28.04 | 26.85 | 34.66 |
| | ZSC | 33.10 | **37.11** | 35.48 | 30.79 | 28.35 | 33.77 |
| | ZS | 23.69 | **25.22** | 24.22 | 22.72 | 23.97 | 22.31 |
| GPT-3 | 5-shot | 24.35 | 24.91 | 24.47 | 22.53 | **26.28** | 23.57 |
| (text-davinci-002) | ZSC | 26.21 | **28.79** | 31.23 | 23.29 | 22.34 | 25.39 |
| | ZS | 33.53 | **36.23** | 34.86 | 29.85 | 30.48 | 36.23 |
| ChatGPT | 5-shot | 34.13 | **37.98** | 36.42 | 31.16 | 31.98 | 33.08 |
| | ZSC | 42.22 | **46.37** | 42.80 | 38.86 | 39.74 | 43.35 |
| | ZS | 53.32 | **61.20** | 58.10 | 45.66 | 48.20 | 53.43 |
| GPT-4 | 5-shot | 55.98 | **63.45** | 60.32 | 48.12 | 51.23 | 56.77 |
| | ZSC | **67.19** | **75.09** | **70.23** | **60.45** | **62.45** | **67.74** |
| Random Guess | | 25.13 | | | | | |

Table 4: Accuracy (%) on Multilingual Compositional Relation (MCR) benchmark on state-of-the-art LLMs via zero-shot (ZS) and zero-shot chain-of-thought (ZSC) prompting settings.

instruction of *"Let's think step by step"*, GPT-4 overall achieves a significant 13.9% increase in accuracy, demostrating strong evidence for the effectiveness of CoT. By enabling in-context learning with five examples in the same language as the questions, few-shot (FS) steers models for 1-2% better performance than zero-shot as shown in Table 4.

However, Llama2 7B and GPT-3 exhibited somewhat anomalous behavior; their performance was on par with random guessing across prompt settings. This behavior indicates the model does not understand the question or answer clearly. We conducted a detailed examination of GPT-3's outputs and found that under ZSC setting, GPT-3 is likely to make illogical inferences, which could be one of the reasons for its low accuracy. This also indicates that while the chain of thought approach can enhance performance by encouraging models to "think through" problems step by step, the effectiveness of this strategy varies significantly across models. The same observation applies to the few-shot prompting technique.

**Effect of Language**    In examining the performance of various LLMs, it becomes evident that language exerts a considerable influence on accuracy. Across the different models and prompt settings, in English, models consistently achieve the highest accuracy in reasoning questions, with a few outliers like Llama2 13B ZS and ZSC, showing marginally better performance in Japanese (JA) and French(FR) respectively.

Languages like Japanese and Korean, to a limited extent, pose challenges to LLMs to achieve comparable performance. This trend is largely anticipated, considering the dominance of English-language content within the training datasets of these models. This prevalence inherently translates into enhanced performance on tasks conducted in English, as compared to those in other languages. It is important to highlight, despite the fact that English (EN) and Chinese (ZH) questions are human-rewritten rather than machine-translated, models are not consistently high-performing in Chinese compared to other non-English languages. In numerous instances, it is found to be inferior compared to French (FR) in different models under various prompt settings.

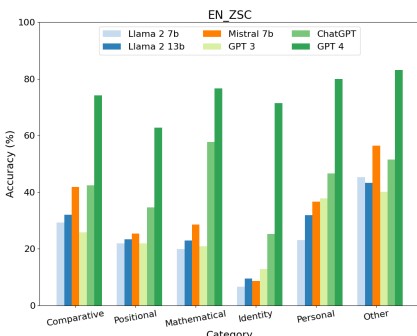

Figure 2: Category breakdown accuracy in English zero-shot cot (ZSC).

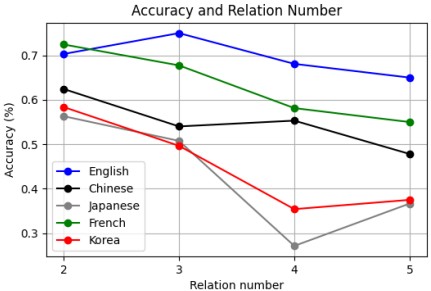

Figure 3: Accuracy(%) regarding #Relations across languages in GPT-4 using zero-shot (ZS).

| ChatGPT | | | |
|---|---|---|---|
| | | T | F |
| GPT 4 | T | 34.97% | 32.77% |
| | F | 6.05% | 26.21% |

| GPT 3 | | | |
|---|---|---|---|
| | | T | F |
| GPT 4 | T | 20.16% | 47.57% |
| | F | 5.23% | 27.03% |

| Mistral 7B | | | |
|---|---|---|---|
| | | T | F |
| GPT 4 | T | 26.09% | 41.65% |
| | F | 7.69% | 24.57% |

| Llama 13B | | | |
|---|---|---|---|
| | | T | F |
| GPT 4 | T | 22.75% | 44.99% |
| | F | 6.43% | 25.83% |

| Llama 7B | | | |
|---|---|---|---|
| | | T | F |
| GPT 4 | T | 15.88% | 51.86% |
| | F | 9.51% | 22.75% |

Table 5: Accuracy of Models against GPT 4 in Chinese (ZH) on the same questions under ZSC.

## 5.3 Breakdown Results

We have conducted a summary and statistical analysis of each language, each prompt, and each model. Figure 2 displays the breakdown performance of all models in six categories in English using zero-shot cot prompting (EN-ZSC).

In the GPT families, we observe that GPT-4 outperforms the rest of the models, and it shows the highest accuracy and averaged 75.1% in all categories, setting it apart from the rest of the candidates. ChatGPT (46.4%) and Mistral 7B (37.1%) show commendable performance with a competitive accuracy across different categories. Explicitly, GPT-4 shows its weakest performance in personal relation reasoning, whereas the rest of the models exhibit their poorest results in identity reasoning. It is worth noting that all models achieve the highest accuracy in the *"other"* reasoning category, except ChatGPT. Appendix C details all five languages in ZS, FS and ZSC prompting methods.

## 5.4 Multilingual Exemplar

We carried out an ablation study to investigate the impact of exemplar languages on reasoning abilities. We employed ChatGPT with a 5-shot prompt methodology, distinguishing between three types of exemplars: **Native**, **Multilingual** and **English**. In the Native Exemplar condition, all five examples were presented in the same language as the question posed. The Multilingual Exemplar condition incorporates exactly one ex-

| Language | Native | Multilingual | English |
|---|---|---|---|
| EN | **37.98** | 36.17 | 37.98 |
| FR | **36.42** | 35.11 | 34.23 |
| JA | 31.16 | **31.56** | 29.18 |
| KO | **31.98** | 31.73 | 29.16 |
| ZH | **33.08** | 31.51 | 31.25 |

Table 6: Accuracy(%) of ChatGPT on 5-shot multilingual, native, English Exemplar.

ample from each of the five distinct languages. For the English Exemplar setting, all examples are in English. The outcomes of this comparison are depicted in Table 6.

ChatGPT attains marginally higher reasoning accuracy in native exemplars than in multilingual exemplars; one exception is that Japanese (JA) is the only language where multilingual exemplars contribute to a slight accuracy improvement. All English exemplar results in worse accuracy than native exemplars. These suggest that reasoning in a target language benefits from examples in the exact same language. There is no obvious correlation between the multilingual exemplar and the English exemplar. One potential reason could be that the multilingual exemplar also contains exactly one native example.

### 5.5 Impact of Relation Numbers

The complexity of a question is proportional to the number of relations shown in a question, posing more challenging tasks for LLMs to process. Figure 3 illustrates the relationship between the number of relations and the accuracy percentage for GPT-4 across different languages when using a zero-shot approach. The data indicates a general decline in accuracy as the relation count grows, with English experiencing the least impact, showing a decrease of only about 5%. In contrast, all other languages assessed exhibit a more marked reduction in accuracy, surpassing a 10% drop as the relations increase. This trend underscores the varying degrees of difficulty that LLMs may encounter as the relation count changes. For other prompting methods and other LLMs, please refer to Appendix D.

### 5.6 Detailed Accuracy Comparison between Models

Table 5 compares the accuracy of five models – ChatGPT, GPT3, Mistral 7B, Llama 13B, and Llama 7B – against GPT 4 across MCR under ZH ZSC setting. If both models answer the same questions correctly (or incorrectly), then models agree with each other; otherwise, they disagree. We observe that ChatGPT and GPT-4 tend to have the highest level of agreement (34.97% + 26.21%). The proportion of both correct and incorrect is the highest among models. Conversely, the agreement between LLaMA-7B and GPT-4 is the lowest, and they disagree with over 60% of the questions. The observation indicates that ChatGPT and GPT-4 may share the most similarity in terms of model architecture, training data, training methods and etc.

## 6 Conclusion and Discussion

In this work, we introduce a Multilingual Compositional Relation (MCR) Benchmark and conduct a comprehensive evaluation of six state-of-the-art LLMs capabilities in reasoning compositional relations. Our findings reveal that the LLMs, including LlamA-2 7B and GPT-3, struggle to navigate through complex compositional relation questions, with their performance in English marginally exceeding that of random guess. This highlights a critical limitation in the current generation of LLMs, suggesting a considerable gap remains before these models can truly understand the semantics of human languages. However, it is noteworthy that Mistral 7B, ChatGPT, and the more advanced GPT-4 exhibit improved accuracy, indicating progressive enhancements in the field. Identifying and understanding these limitations is crucial for improving the model's performance. Investigating the underlying causes of these deficiencies could lead to advancements in natural language understanding, model architecture, and training methodologies.

## Limitation

We investigate a subset of publicly available LLMs. For GPT models, our evaluation only exterminated a representative checkpoint. The current MCR covers five indicative, distinct languages in total and the evaluations show consistent results; nevertheless, we are going to expand the language coverage in the future.

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

## A Motivation

### A.1 Family Relation Reasoning Query Example

See Table 7.

| **Example: Relation Number = 1, EN** |
|---|
| {A}'s parent is {B}, {B}'s child is? |
| A) {A} |
| B) {B} |
| C) Neither |
| D) Uncertain |
| Answer: A |

| **Example: Relation Number = 1, ZH** |
|---|
| {A}的家长是{B}, {B}的孩子是? |
| A) {A} |
| B) {B} |
| C) 都不是(Neither) |
| D) 不确定(Uncertain) |
| Answer: A |

| **Example: Relation Number = 2, EN** |
|---|
| {B}'s dad is {A}, {A}'s brother{C}, {C}'s nephew is? |
| A) {A} |
| B) {B} |
| C) {C} |
| D) Neither |
| E) Uncertain |
| Answer: B |

| **Example: Relation Number = 2, ZH** |
|---|
| {B}的爸爸是{A}, {A}的哥哥是{C}, {C}的侄子是? |
| A) {A} |
| B) {B} |
| C) {C} |
| D) 都不是(Neither) |
| E) 不确定(Uncertain) |
| Answer: B |

| **Example: Relation Number = 3, EN** |
|---|
| {A}'s wife is {B}, {B}'s son is {C}, {C}'s son is {D}, {D}'s grandpa is? |
| A) {A} |
| B) {B} |
| C) {C} |
| D) {D} |
| E) Neither |
| F) Uncertain |
| Answer: A |

| **Example: Relation Number = 3, ZH** |
|---|
| {A}的老婆是{B}, {B}的儿子是{C}, {C}的儿子是{D}, {D}的爷爷是? |
| A) {A} |
| B) {B} |
| C) {C} |
| D) {D} |
| E) 都不是(Neither) |
| F) 不确定(Uncertain) |
| Answer: A |

Table 7: Reasoning Question Examples. Examples show up to three relation numbers in English (EN) and Chinese (ZH)

## A.2 Definition Relation Example

See Table 8

| **Example: Relation Number = 1, EN** |
| --- |
| Am I my mother's child? |
| A) Yes |
| B) No |
| C) Uncertain |
| Answer: A |

| **Example: Relation Number = 1, ZH** |
| --- |
| 我是我妈妈的孩子吗? |
| A) 是 |
| B) 不是 |
| C) 不确定 |
| Answer: A |

| **Example: Relation Number = 2, EN** |
| --- |
| Is my brother's son my nephew? |
| A) Yes |
| B) No |
| C) Uncertain |
| Answer: A |

| **Example: Relation Number = 2, ZH** |
| --- |
| 我弟弟的儿子是我侄子吗? |
| A) 是 |
| B) 不是 |
| C) 不确定 |
| Answer: A |

| **Example: Relation Number = 3, EN** |
| --- |
| Is my dad's mom's husband my grandpa? |
| A) Yes |
| B) No |
| C) Uncertain |
| Answer: A |

| **Example: Relation Number = 3, ZH** |
| --- |
| 我爸爸的妈妈的老公的是我爷爷吗? |
| A) 是 |
| B) 不是 |
| C) 不确定 |
| Answer: A |

Table 8: Reasoning Question Examples. Examples show up to three relation numbers in English (EN) and Chinese (ZH)

# B   Sample Questions for Each Category

See Table 9.

| 1 | Max's car is more expensive than John's car, and Andy's car is cheaper than John's car. How does Max's car compare to Andy's car in terms of price? Answer choices: A) More expensive B) Cheaper C) Uncertain |
|---|---|
| 2 | The telephone and the television were invented in the same year. The electric light was invented two years after the telephone, and the telephone was invented three years before the mobile phone. How does the mobile phone compare to the electric light in terms of invention time? Answer choices: A) Invented two years earlier B) Invented one year later C) The same time D) Uncertain |
| 3 | Location A is colder than Location B, and Location C is warmer than Location A. Is Location C warmer or colder than Location B? Answer choices: A) Warmer B) Colder C) Uncertain |

(a) Comparative

| 1 | The mall is southwest of Frank's home, and the post office is west of the mall. In what direction is the post office from Frank's home? Answer choices: A) Northeast B) Southeast C) Southwest D) Northwest |
|---|---|
| 2 | Robert is 300 meters northeast of the teaching building, and the teaching building is 300 meters southwest of Lucy. Where is Lucy in relation to Robert? Answer choices: A) Northeast B) Same position C) Southwest D) Uncertain |
| 3 | Lee sits to the left of Williams, and Zhang sits in front of Williams. In which direction is Zhang from Lee? Answer choices: A) Front left B) Left C) In front D) Front right |

(b) Positional

| 1 | Rosie is 21 years old. Rosie's mother's age is twice that of Rosie. How old will Rosie's mother be next year? Answer choices: A) 43 years old B) 42 years old C) 21 years old D) Uncertain |
|---|---|
| 2 | In a plane, Line AB passes through point O, and Line CD passes through point P. What is the relationship between Line AB and Line CD? Answer choices: A) Intersecting B) Perpendicular C) Parallel D) Uncertain |
| 3 | Jack's time to run 100 meters is twice that of Lee's, and Chris's speed is twice that of Lee's. Who is the fastest? Answer choices: A) Jack B) Lee C) Chris D) Uncertain |

(c) Mathematical

Table 9: Sample Questions

| 1 | Circles A, B, and C are on a straight line. Point A is 4 centimeters to the left of Point B, and Point B is 4 centimeters to the right of Point C. All of them have a diameter of 2 centimeters. What is the relationship between Circle A and Circle C? Answer choices: A) Completely coincide B) Separated C) Containment relationship D) Intersect |
|---|---|
| 2 | Rosie is 8 years old. Mike is 2 years younger than Rosie. Joe is 2 years older than Mike. How old is Mike this year? Answer choices: A) 6 years old B) 8 years old C) 10 years old D) 12 years old E) Uncertain |
| 3 | On a multi-tier bookshelf, 'A Brief History of Time' is on the top tier and 'The Great Gatsby' is on the tier right below 'A Brief History of Time'. 'One Hundred Years of Solitude' is on the tier right above 'The Great Gatsby'. Where is 'One Hundred Years of Solitude' in relation to 'A Brief History of Time'? Answer choices: A) Upper tier B) Lower tier C) Same tier D) Adjacent E) Uncertain |

(d) Identity

| 1 | Teacher Zhang is Teacher Wang's math teacher, and Teacher Wang is Student Lee's Chinese language teacher. Who is Student Lee's math teacher? Answer choices: A) Teacher Zhang B) Teacher Wang C) Neither D) Uncertain |
|---|---|
| 2 | Wang is an employee of Company A, and Lee is the CEO of Company A. Wang's CEO is Ming. What is Lee to Ming? Answer choices: A) The same person B) CEO C) Employee D) Uncertain |
| 3 | Mike is William's son, Bai is Mike's daughter, Mary is William's wife. Mary is Bai's? Answer choices: A) Wife B) Mother C) Uncertain D) Grandmother |

(e) Personal

| 1 | Event A occurs only if Event B happens, and Event B occurs only if Event C happens. Event C leads to Event D. Event A did not happen. Does Event D happen? Answer choices: A) Yes B) No C) Uncertain |
|---|---|
| 2 | iPhone and Galaxy are the same type of product, iPhone and iPad are different types of products. What is the relationship between Galaxy and iPad? Answer choices: A) Same type of product B) Different types of products C) Uncertain |
| 3 | Event B occurs only if Event A happens, and Event C occurs only if Event B happens. Event A did not happen. Event C occur? Answer choices: A) Yes B) No C) Uncertain |

(f) Other

Table 9: Sample Questions (Continued)

## C   Breakdown Results

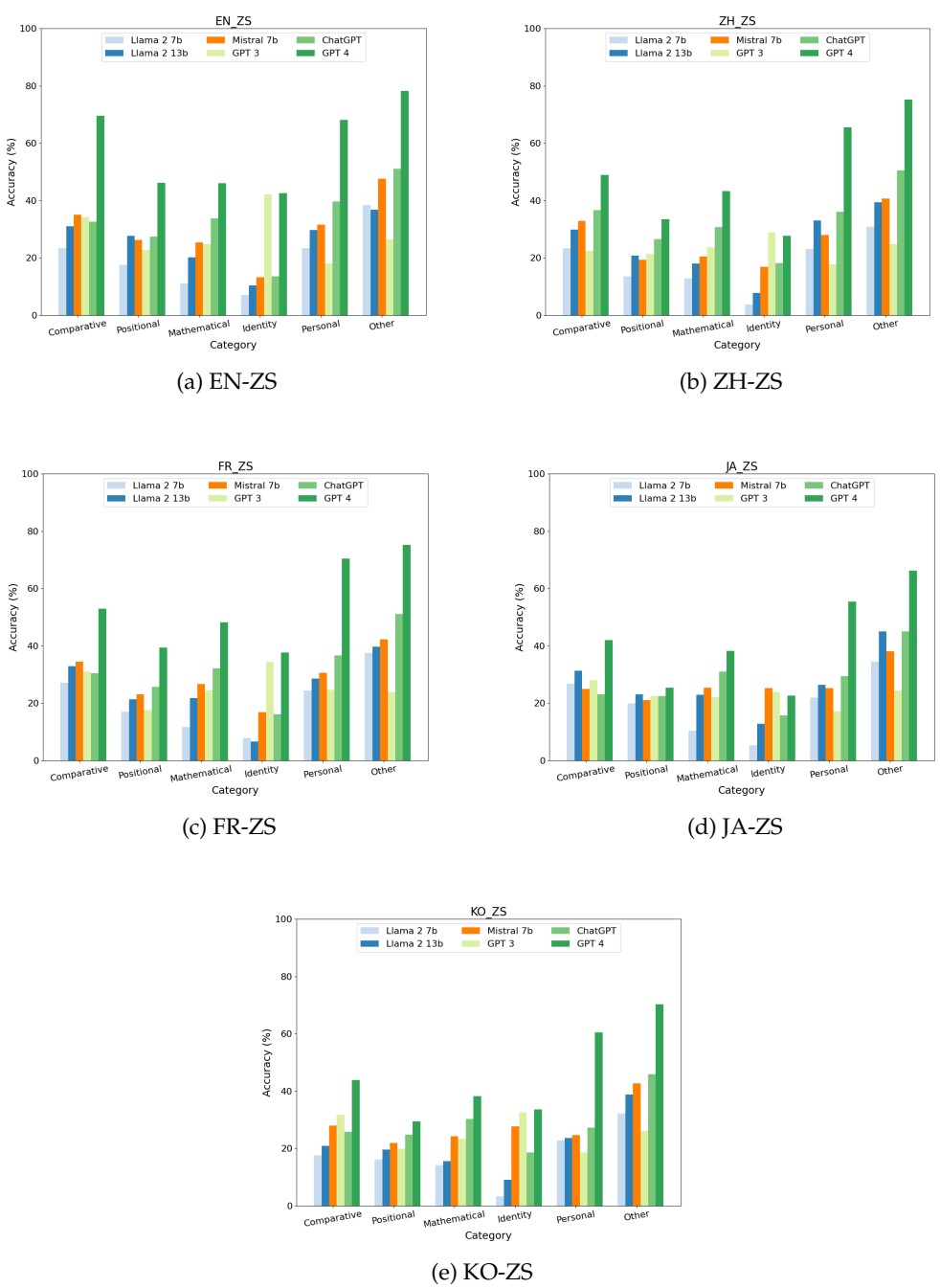

Figure 4: Category breakdown accuracy, Zero-shot (ZS) prompt

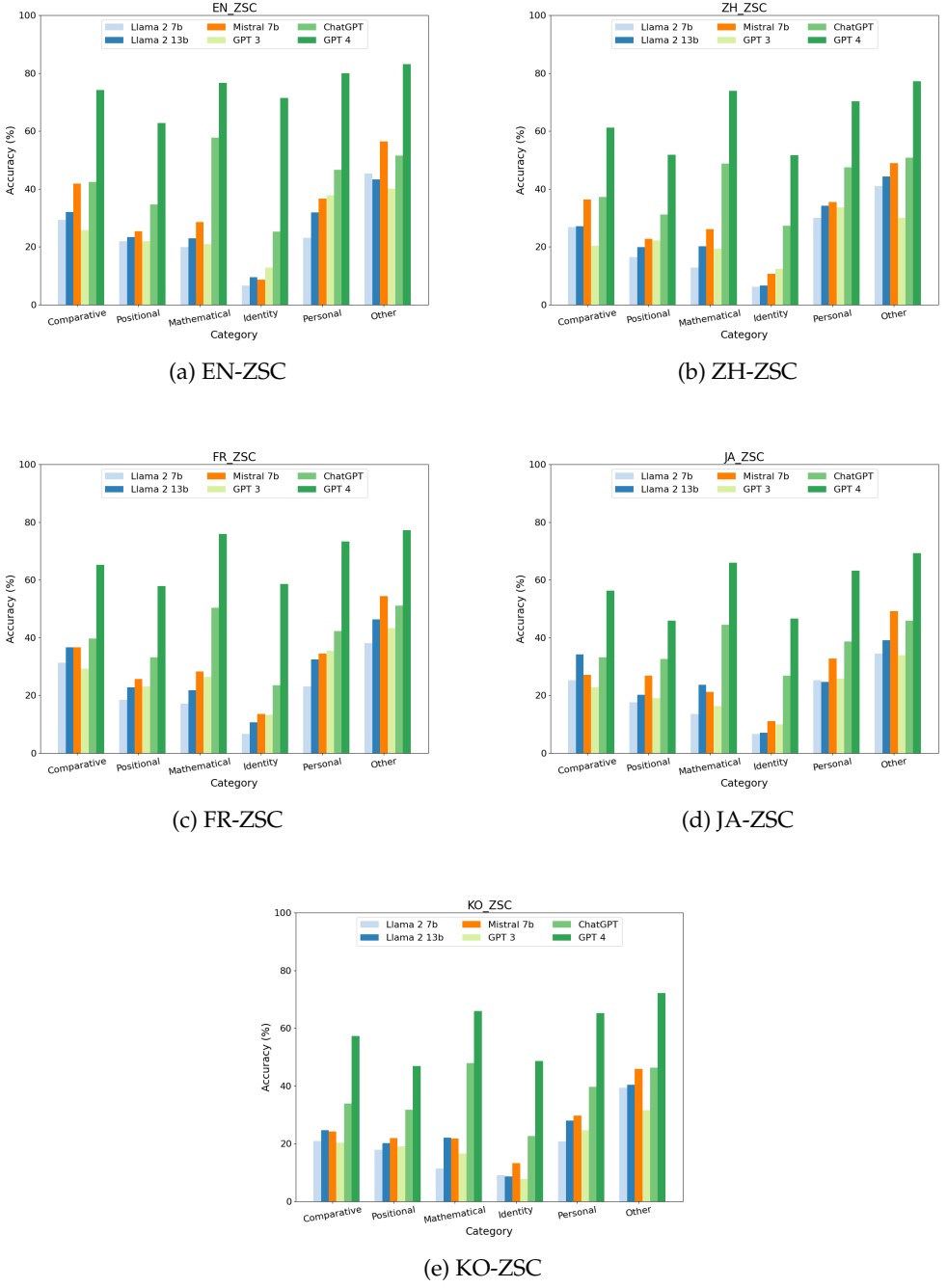

Figure 5: Category breakdown accuracy, Zero-shot Chain-of-Thought (ZSC) prompt

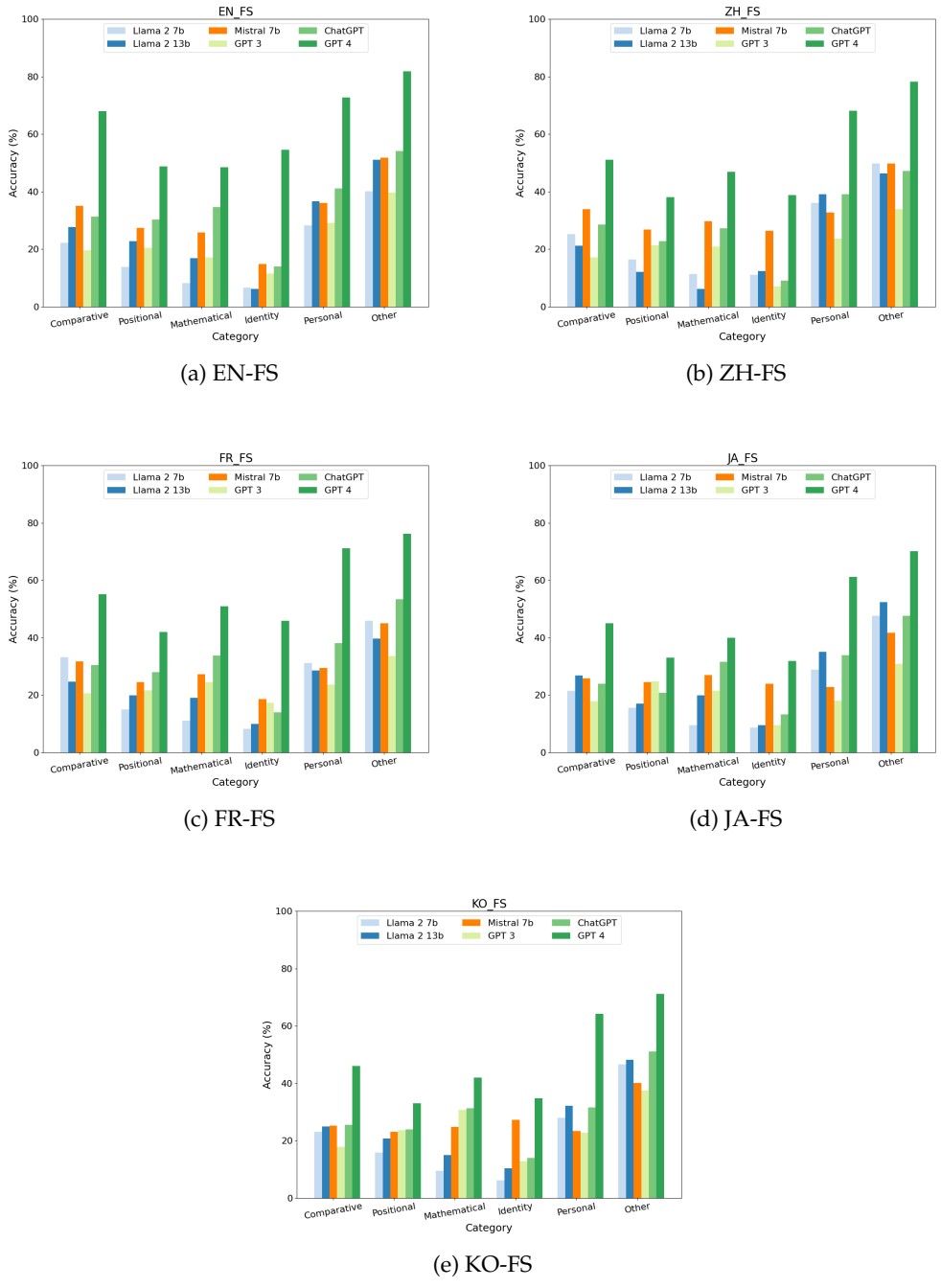

Figure 6: Category breakdown accuracy, Few-shot (FS) prompt

# D   Impact of Relation Number

See Figure 7.

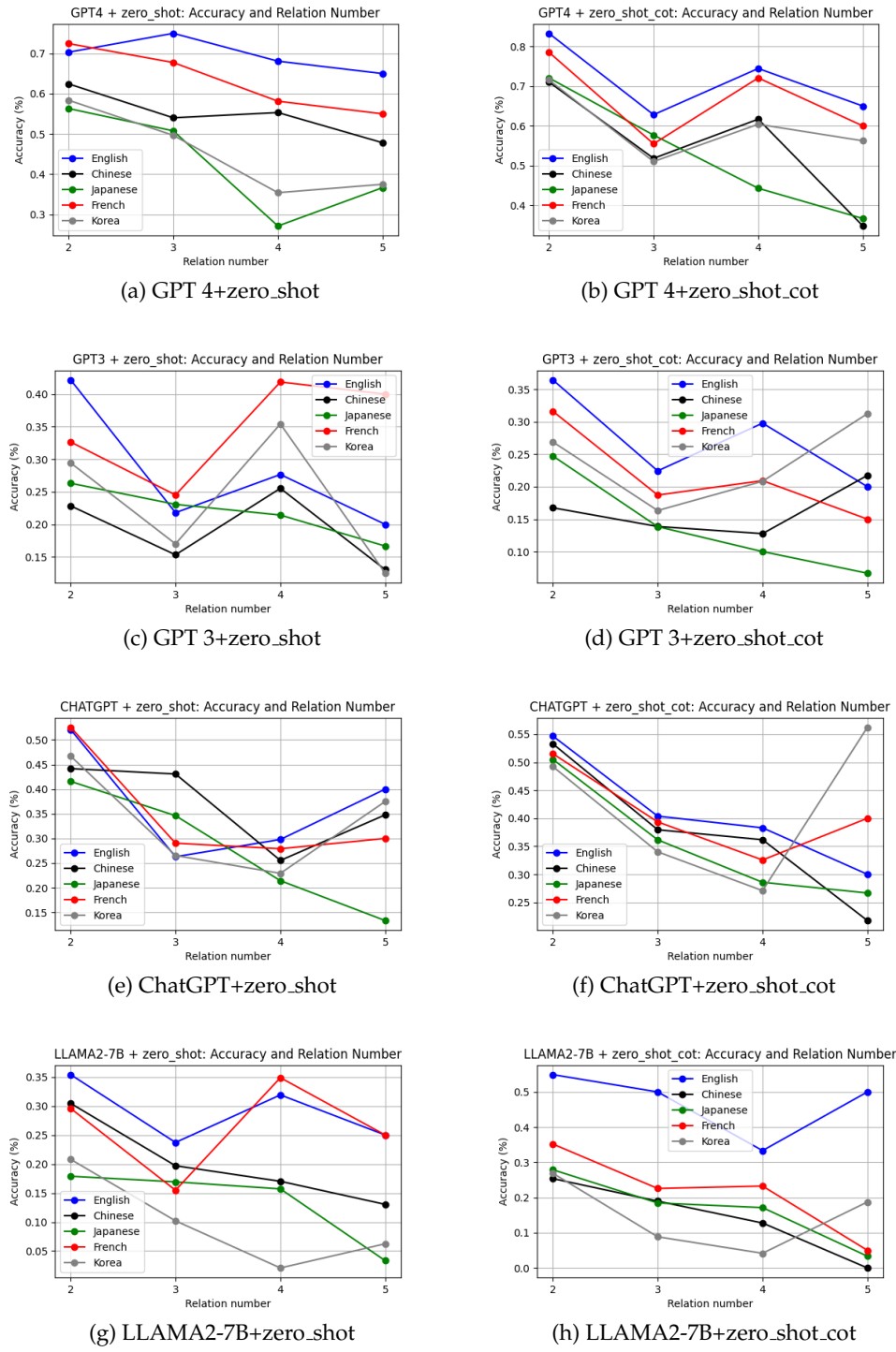

Figure 7: Relation number impact. There is no as clear decreasing trend in GPT3 plots when the relation number increases. We believe it is because GPT3's accuracy is already close to the probability of random guess.

