# OpenReview forum: "Large Language Model is not a (Multilingual) Compositional Relation Reasoner"
_colmweb.org/COLM/2024/Conference — COLM_

### Official Review · Reviewer_RFfM · 2024-05-11

**Rating:** 7
**Confidence:** 4
**Ethics Flag:** 1

**Summary:**

This paper introduces a multilingual dataset with 1800 examples of various compositional relations. It presents a set of experiments to test the performances of LLMs (open-sourced and not) in this specific reasoning task. The dataset is interesting, and the paper is not bad; the authors also speculate about the results that would foster future research in this direction. However, the paper is badly written. It is still understandable, but it would require generous editing from a native speaker. I know that this is a sufficient reason to reject a paper at major NLP conferences. Since this is a new conference, I would like to start a new trend: accept this paper and strongly suggest giving them a better dress with the help of a professional editor.

Here my comments:

English should be carefully revised, there are several statements hard to process (e.g. "our evaluation only exterminated a representative checkpoint"). Also, textual flow is not optimal and in several paragraphs the text sounds like broken syntax (e.g. in the introduction).

I would make a strong distinction in the future, both at the level of the dataset and experimentally, between lexicalised relations that involve commonsense or semantic knowledge (e.g. kinship terms) and other relations that would be treatable even if you put dummy words in the context. Indeed it would be very interesting to see results when you reformulate for example a comparative or identity relation with nonsense nouns. If the performance are the same it means that the inference is triggered by the syntax (and I guess it is in a lot of cases). It would make sense to include examples more complex in the dataset with a convoluted syntax.

With careful inspection, the category "Other" can be split into more meaningful sub-categories. The better performance of all the LLMs for this category could be due to a subset of examples that are very easy to process.

Concerning the zero shot CoT scenario, I need to see concrete examples of the dialogical interaction and how long it takes to arrive at the right solution.

**Questions To Authors:**

Why Japanese and Korean in Figure 3 show better performance for 5 than for 4 relations? It is counter-intuitive. Could it be something related to the syntax of these languages?

**Reasons To Accept:**

Interesting dataset
Interesting task at the intersection between syntax and semantics

**Reasons To Reject:**

Badly written, it needs professional editing

---

> ### Author Rebuttal · Authors · 2024-05-30
>
> We sincerely appreciate your dedication to reviewing our manuscript thoroughly. Your constructive comments and valuable expertise have greatly contributed to the improvement of our paper. We will engage professional editing services to refine the manuscript before submitting the final version.
>
> The perspective on syntax that you mentioned presents an intriguing avenue for future work. It may also relate to the counterintuitive phenomena associated with Japanese that you highlighted later in your comments.
>
> __question__
>
> The syntax of Japanese is certainly a potential reason for the observed phenomenon, given its grammatical differences from other languages. Other factors may also contribute to this anomaly, such as：
> 1. The performance of the translator. We translated the questions from English to Japanese, and the significant grammatical differences between these languages could have led to some unexpected interactions.
> 2. Whether the pre-trained data contains similar relational structures is unknown since GPT4 is not open-source.

---

> ### Comment · Reviewer_RFfM · 2024-06-05
> **no major concerns**
>
> I went through all the reviews and comments, from my side the paper is ok and it can be accepted.

---

### Official Review · Reviewer_qSX2 · 2024-05-12

**Rating:** 6
**Confidence:** 3
**Ethics Flag:** 1

**Summary:**

This paper presents a dataset of multilingual compositional reasoning. The dataset considers a variety of  compositional relationships: such as Positional, Comparative, Personal, Mathematical, Identity. The dataset, MCR, is used to evaluate many popular large language models (GPT4, ChatGPT, Llama, Mistral, etc. Three prompting techniques are used.

**Questions To Authors:**

* Can you say more about where error cases occur? Give more examples of language models making mistakes on this data?
* Why is only 5-shot available for ChatGPT?

**Reasons To Accept:**

Understanding compositionally of language models is a very important problem. This paper introduces datasets and to better understand where language models make mistakes in compositional reasoning.

The dataset is of reasonable size, the kinds of compositional reasoning selected seem quite reasonable. The dataset appears to be challenging  enough to demand further study for models of a wide variety of sizes (7B to GPT4 sized).

**Reasons To Reject:**

It is clear from the empirical results that language model performance is not up to expectations on these compositional. While the authors break down the performance of the models in terms of relation numbers, prompting strategy, model size, exemplars, I feel that the paper would be much stronger if it further investigated the performance of the models in ways that may provide researchers with (1) greater insights into the challenges of compositional reasoning (2) ideas for mitigations/improvements to compositional reasoning systems.

For instance I am curious about the following:
* Do models tend to get the same examples correct or incorrect?
* Why is decoding mode used for these models in non-chain of thought generations — why not scoring mode? This way you could also look at uncertainty/perplexity of answer? And correlate uncertainty on particular questions across models?
* Do the certain kinds of entities used in the questions affect performance? E.g. when person names are used are models better or worse  than when entities are Events A, B, C?
* How much could prompt engineering improve results?
* Why are there only five-shot results for ChatGPT?

I would personally like to understand where models make mistakes in this data. Some of the questions I think are quite tricky:

* Is my dad’s mom’s husband my grandpa? -> this question’s answer makes some assumptions. For instance, there is a universe where the dad’s mom has remarried and the question asker doesn’t consider this husband their grandfather.
* Williams participates in the Miss Universe contest, one of the selection criteria for which is being enthusiastic. What kind of person is Williams? -> It is not clear if the selection criteria refers to what is needed to be a contest or to with the contest. And so it is not clear how to answer this question in my opinion.

Other
* humongous is a word that in my opinion is too informal to use in a research paper

---

> ### Author Rebuttal · Authors · 2024-05-30
>
> Thank you very much for taking the time to read and review our manuscript. Below is our response.
>
> __Why does MCR pose a challenge to LLMs?__
>
> We believe it is a challenge because firstly, similarly to LLM’s reversal curse, current models are essentially a next-token generator/predictor. Asking them to reason reversally through contexts is challenging. Second, in MCR benchmark questions, the correct answer may not be a word seen in the question. Eg, mom’s dad is grandpa, the negation of large is small, etc.
>
>
> __Do models tend to get the same examples correct or incorrect?__
>
> We employ greedy decoder sampling by setting the temperature to 0. If allowed by the API, we have also set top_k to 1.
>
> __why not scoring mode?__
>
>  3 out of 6 models in the experiment are closed-sourced and only accessible via API, making it difficult to directly access the models’ outputs before decoding.
>
> __How much could prompt engineering improve results?__
>
> In the paper, chain-of-thought and few-shot prompting are used. We keep the settings the same as the original paper to showcase the challenge in reasoning through MCR. We will continue to explore such limitations and how to improve correctness via prompting / fine tuning in the future work.
>
> __Is my dad’s mom’s husband my grandpa?__
>
> This is a potential limitation and we will incorporate explanations in the according section. Some questions in the MCR benchmark include certain assumptions. When querying models, we found they also have some assumptions. Eg, in a definition query “what’s dad’s mom’s husband?”, the model outputs “grandpa”. Such assumptions could come from the training data.
>
> __Why are there only five-shot results for ChatGPT?__
>
> After conducting experiments with few-shot learning on ChatGPT, we observed that the improvements were significantly smaller compared to zero-shot CoT (Chain of Thought). This suggests that the importance of exemplars may be less critical than CoT for these types of problems.
>
> __Can you say more about where error cases occur?__
>
> There is a significant disparity in accuracy across different types of questions. For instance, in identity problems such as "A is 100 meters to the left of B, and B is 100 meters to the right of C, where is A in relation to C?", the model struggles to recognize that A and C are in the same position. Additionally, using ChatGPT as an example, it tends to answer "100 meters to the right." which is substring seen in the question.

---

> > ### Comment · Reviewer_qSX2 · 2024-06-02
> >
> > thank you for your response. a few follow ups:
> >
> > >Do models tend to get the same examples correct or incorrect?
> >
> > I mean to ask among the experiments you ran on your datasets did the models you evaluated get the same examples correct? the same ones incorrect? e.g. pick two models and see counts of {Both Correct, Both Incorrect, Model 1 correct only, Model 2 correct only}. Is this more clear?
> >
> > >Why are there only five-shot results for ChatGPT?
> >
> > I mean to ask -- am I missing five shot results with other models?
> >
> > >Can you say more about where error cases occur?
> >
> > Can you say more? Are there any common trends I should be thinking of beyond what is mentioned in the paper?
> >
> > >Is my dad’s mom’s husband my grandpa?
> >
> > I guess my suggestion would be to assess a subset of questions to determine the percentage that are unambiguous.

---

> > ### Author Response · Authors · 2024-06-04
> > **Response**
> >
> > Thanks for your explanation, below is our response.
> >
> > __Why are there only five-shot results for ChatGPT?__
> >
> > We rerun the 5-shot experiments on all models on English(EN) and Chinese(ZH).
> > |   model  |   prompt |    EN    |   ZH  |
> > |-----------|------------|----------|------|
> > | Llama2 7B|  ZS       |    30.1   |  24.8 |
> > |   |   5-shot |    30.64  |  24.7|
> > |   |      ZSC  |    35.2 | 24.5|
> > | Llama2 13B|  ZS       |    38.8   | 31.9|
> > |   |   5-shot |    38.0  |  32.2|
> > |   |      ZSC  |    36.4 | 33.0|
> > | Mistral 7B|  ZS       |  34.9 | 40.1   |
> > |   |   5-shot |    36.87  |  40.84 |
> > |   |      ZSC  |    47.5 | 40.3|
> > | GPT-3|  ZS       |    31.8   |  20.0
> > |   |   5-shot |    25.78  |  31.44|
> > |   |      ZSC  |    29.6 |  15.6 |
> > | ChatGPT|  ZS       |    39.3   | 41.1|
> > |   |   5-shot |    40.5  |   40.1 |
> > |   |      ZSC  |    46.5 | 44.3|
> > | GPT-4|  ZS       |    71.6   | 58.0 |
> > |   |   5-shot |    74.7  |  65.84  |
> > |   |      ZSC  |    75.2 |  61.4 |
> >
> > __Do models tend to get the same examples correct or incorrect?__
> >
> > The answer varies based on the model we pick. Use Chinese ZSC as an example, below are the statistics we have to compare GPT-4 as 3 other models, all the numbers are in percentage:
> >
> > 1. GPT-4 vs ChatGPT
> > |  | | ChatGPT| |
> > |--|-|--------|-----------|
> > |   | |correct| incorrect|
> > |GPT-4|correct|39.4|22|
> > | |incorrect|4.9|33.7|
> >
> > 2. GPT-4 vs Llama2 7B
> > |  | | Llama2 7B| |
> > |--|-|--------|-----------|
> > |   | |correct| incorrect|
> > |GPT-4|correct|16.6|44.8|
> > | |incorrect|3.7|34.9|
> >
> > 3. GPT-4 vs Mistral 7B
> > |  | | Mistral 7B| |
> > |--|-|--------|-----------|
> > |   | |correct| incorrect|
> > |GPT-4|correct|25.7|35.6|
> > | |incorrect|14.6|24|
> >
> > We find that although ChatGPT and Mistral 7b have very close performances (40.3/44.3 for ZH ZSC), they tend to have different distributions in this experiment. ChatGPT has higher both correct and incorrect compared with Mistral 7B.  This might be due to the difference in model architecture, pre-train/finetuning data, pre-train method, etc.
> >
> > __Can you say more about where error cases occur?__
> >
> > Below are our conclusions:
> >
> > __1__. We observed a pattern in the errors present in the benchmark. When the symbols connecting two relations are identical, LLMs are less likely to make mistakes. For example, if A is greater than B, and B is greater than C, the LLM can easily deduce that A is greater than C.
> >
> > However,  use ChatGPT as an example, if we change one relation symbol in the question:
> >
> > "if A is greater than B, C is smaller than B,  and C is greater than D, then A is __ than D.
> >
> > A) greater
> >
> > B) smaller
> >
> > C) Uncertain"
> >
> > The thought its output would be
> >
> > "If A is greater than B, and C is smaller than B but greater than D, we cannot definitively determine the relationship between A and D. It depends on the specific values of A, B, C, and D. So, the correct answer would be:
> > C. Uncertain"
> >
> > which is incorrect.
> >
> > This might also be the reason that for the question "A is 100 meters to the left of B, and B is 100 meters to the right of C, where is A in relation to C?", ChatGPT chooses the incorrect answer"100 meters to the right.".
> >
> > __2__. When the symbols connecting two relations are identical, but the relation between two asking objects is uncertain, the   LLM tends to answer the relation it seen in the question, for example:
> >
> > "The hospital is west of the Supermarket, and the school is west of the supermarket. In which direction is the hospital from the school?
> >
> > A. the hospital is west of the school
> >
> > B.  the hospital is east of the school
> >
> > C. Uncertain"
> >
> > ChatGPT outputs "Given that both the hospital and the school are located to the west of the supermarket, we can conclude that the hospital is west of the school as well. So, the correct answer is:
> >
> > A. the hospital is west of the school"
> >
> > where the correct answer should be "Uncertain".
> >
> > __3__ We noticed that how the question being asked also matters.
> >
> > If we ask ChatGPT:
> >
> > "Mike is Amy's father; John is Mike's father; then John is Amy's?
> >
> > A.  father
> >
> > B. grandfather
> >
> > C. son
> >
> > D. uncertain"
> >
> > in this question, John is the subject in both the relation we provided and the question, ChatGPT outputs
> >
> > "If John is Mike's father and Mike is Amy's father, then John is Amy's grandfather. So, the correct answer is:
> >
> > B. grandfather" which is the correct answer.
> >
> > But if we change the prompt to
> >
> > "Mike is Amy's father; John is Mike's father; then Amy is John's?
> >
> > A.  daughter
> >
> > B. grandchild
> >
> > C. mother
> >
> > D. uncertain"
> >
> > The output becomes
> >
> > "Amy is John's daughter. So, the correct answer is:
> >
> > A. daughter" which is clearly a wrong answer.
> >
> > __Is my dad’s mom’s husband my grandpa?__
> >
> > Thanks again for your suggestion. We will re-examine the questions and remove these ambiguous questions from the final version.

---

> > > ### Comment · Reviewer_qSX2 · 2024-06-05
> > >
> > > Thank you so much. This kind of analysis I believe greatly strengthens the paper. I would recommend adding all of these results to the paper and even doing more of such analysis.

---

### Official Review · Reviewer_rZGM · 2024-05-12

**Rating:** 4
**Confidence:** 4
**Ethics Flag:** 1

**Summary:**

This paper proposes a multiple choice QA data set for measuring compositional reasoning capabilities of language models. Several pieces of information, that are e.g. related by transitivity, need to be combined to correctly answer the questions. The data set is translated into a total of 4 languages. The paper contains results for 6 LLMs and two prompting strategies. The data set seems to be challenging for current LMs, the best (GPT-4) shwoing an accuracy of ~75% for the English version of the questions. Despite some of the limitations (see below), data sets like this might attract some attention

===

Update after the author response: The authors did not address my concerns about the data creation provcess and data validation in the slightest, so I update my score to borderline reject.

**Reasons To Accept:**

Clearly written, easy to read

A new data set with questions that are still challenging for current LLMs

Tested on a few LLMs and two prompting strategies

**Reasons To Reject:**

The process of how the questions were created is unclear. I.e. who were the question creators, and what were their guiding principles in creating those questions?


An independent (of question writers) validation of the quality of the data set is missing, and related to that a indicator what the human accuracy on the data set would be.

---

> ### Author Rebuttal · Authors · 2024-05-31
>
> Thank you very much for reading and reviewing our manuscript. Below is our response.
>
> __The process of how the questions were created is unclear.__
>
> The motivation to generate this benchmark is that we believe it is a challenge because:
> 1. Similarly to LLM’s reversal curse, current models are essentially a next-token generator/predictor. Asking them to reason reversally through contexts is challenging.
> 2. In MCR benchmark questions, the correct answer may not be a word seen in the question. Eg, mom’s dad is grandpa, the negation of large is small, etc.
> b
> Our primary focus lies in constructing each relational function, which we then combine to form a problem involving composition relations. And each question does not need to be very long.
>
> Taking our comparative category as an example, each function can follow the same syntactic pattern: "[entity A] is more [adjective] than [entity B]." Thus, each instance might be structured as follows: "[entity A] is more [adjective1] than [entity B], [entity B] is more [adjective2] than [entity C], [entity A] is more ___ than [entity C]" or change the question to "[entity C]  is more ___ than [entity A]?"
>
>
> The answers are human-confirmed. Any issues or ambiguities were adjusted or removed during the process.

---

> > ### Author Response · Authors · 2024-06-04
> > **more explanations**
> >
> > We apologize if our previous explanation was unclear. Allow me to provide a more detailed description.
> >
> > __Creation Procedure__
> >
> > We have employed a group of graduate students, including authors, and divided them into three distinct roles: question creators, answer verifiers and quality checkers.
> >
> > In the question-creating stage, we sourced texts from various online platforms, such as social media and Wikipedia. We gathered simple relationships within these texts. For example, comparative relationships like '[entity A] is more [adj] than [entity B]' were patterns we searched for. We then trained a group of seven workers to integrate different relationships and rephrase them into questions.
> >
> > Following this, we assigned four workers as answer verifiers. Each question was distributed to two workers, and we removed the question if both workers answered incorrectly. This procedure helped us filter out roughly 10% of the initial questions, ensuring a higher level of accuracy.
> >
> > Lastly, we trained two workers to assess the quality of both the questions and their corresponding answers. These workers had access to complete information, focusing particularly on detecting any potential ambiguities in the questions. This final step was crucial in maintaining the integrity and clarity of our question set.
> >
> > __Human Accuracy__
> >
> > We are sorry that we did not conduct extensive human experiments. Based on the results from the answer verifiers, the human accuracy for the remaining questions is approximately 91.2%.
> >
> >
> > Please let me know if you have concerns.

---

### Official Review · Reviewer_C3R4 · 2024-05-12

**Rating:** 7
**Confidence:** 4
**Ethics Flag:** 1

**Summary:**

In this well-written paper, the authors examine the reasoning capabilities of 6 different large language models in terms of up to a 1800 test cases out of the relational realm, i.e. the models need to reason as to whether comparative, identity, mathematical or other relations hold between queried cases. The authors test whether the LLM performance compares across zero- and few-shot scenarios, as well as in chain-of-thought prompting and they also do this in up to 5 languages (English, Chinese, French, Japanese, Korean). This Multilingual Composition Relation benchmark will be published along successful acceptance of the paper.

**Questions To Authors:**

* in the settings of few-shot and/or chain-of-though prompting, has the wording of the prompt itself and the examplars in it varied as well, esp. also in the multilingual cases? this could further point to the robustness of the models
* would there be away to more decisively say that better performance comes with sheer model size, as you seem to allude to with GPT-4 ?

**Reasons To Accept:**

* interesting problem-space, esp. the native and the multilingual settings are something that is rarely assessed with current LLMs
* thorough experimentation and ablation studies
* the authors picked topologically different languages and also ones that potentially have a very different representation within an LLMs pre-training or finetuning

**Reasons To Reject:**

* while the authors to provide analysis of the impact of either the model (size) as such, as well as the method of prompting, consistent performance is probably still not measurable
* since the LLM space moves so fast, there could have been other, newer models that probably should have been test along, esp. ones that are tailored towards instruction following

---

> ### Author Rebuttal · Authors · 2024-05-30
>
> Thank you very much for taking the time to read and review our manuscript. We deeply appreciate the valuable insights and suggestions you have provided.  Below is our response.
>
> __weakness1__
>
> In our experiment, to ensure maximum consistency, we employ greedy decoder sampling by setting the temperature to 0. If allowed by the API, we have also set top_k to 1.
>
> We agree that consistency is a common issue; however, we did not address it in our work for several reasons:
> 1. Space constraints in our manuscript.
> 2. Consistency is not the focus of our study. We employed the zero-shot CoT method, which does not consider consistency in its framework, as evidenced by the absence of this aspect in both the related paper and code.
> 3. As demonstrated in the "Self-consistency" paper (wang et al, 2023), even with the application of self-consistency, most benchmarks show an accuracy improvement of less than 10%. We believe that, aside from models like GPT-4, a 10% improvement does not necessarily indicate strong performance for other models.
>
> __weakness2__
>
> We have selected some of the most prominent and highest-performing models currently available. Thank you for your suggestion. We will continue to update the table as new models are released.
>
> __question1__
>
>
> Verify the robustness of the model by varying the wording of the prompt. This is a commendable idea to test the model's robustness. Currently, our paper primarily focuses on stability from a compositional relation perspective, and in future work, we shall explore the details of all prompts to improve the correctness in the benchmark. For the zero-shot CoT prompting, we aim to align with prior work in settings (Kojim et al,2022). At this stage, enhancing accuracy through prompts is not our primary focus; instead, we have employed classical methods to identify that MCR is the current issue with the LLMs.
>
> __question2__
>
>
> Our experiment covers 6 models, including SOTA and legacy models, to illustrate the inadequacy of compositional relation reasoning. The size of the model is a crucial factor in determining its performance. Other factors may include the data used for training, the model's architecture, and the training methods. This paper utilizes the most popular and representative models currently available. The size of GPT-4 remains uncertain, making it difficult to confirm whether its performance on the MCR benchmark is attributable to its potentially large size.

---

> > ### Comment · Reviewer_C3R4 · 2024-06-02
> >
> > thank you for the additional information, please note some of these in the final version of the paper.

---

### Comment · Area_Chair_Ey7T · 2024-06-03
**Discussion period**

Hi everyone! Thanks to reviewers C3R4 and qSX2 for your reply to author response. For the other reviewers, I'd appreciate it if you could look through the authors' comments soon and let us know if your concerns have been addressed / if there are any outstanding issues soon so there's time to discuss.

With appreciation,
Your AC

---

### Decision · Program_Chairs · 2024-07-10

**Decision:**

Accept

**Comment:**

This paper describes an evaluation dataset for LLMs involving compositional relational reasoning (e.g. "if A is perpendicular to B, and B is perpendicular to C, what is the relationship between A and B"). There are two sets of experiments: the first shows a strong dissociation between the ability to produce relational definitions and to reason relationally about situations (in a kinship domain), the second shows overall poor performance (and significant differences across languages) in a set of six domains.

Several details about the dataset creation and validation process have been added during the discussion period, as well as some discussion of failure modes. I still think the paper would be significantly strengthened by some more in-depth / quantitative analysis of different types of failures, but there seems to be reviewer consensus that it's acceptable in its current form.